# META-TRANSFORMER: A UNIFIED FRAMEWORK FOR MULTIMODAL LEARNING

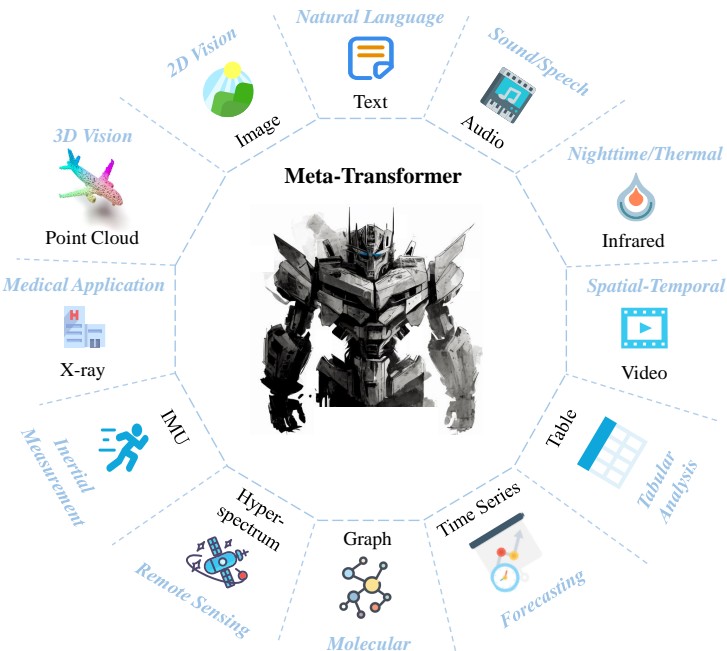

Figure 1: **Unified Multimodal Learning**. Meta-Transformer utilizes the same backbone to encode natural language, image, point cloud, audio, video, infrared, hyperspectral, X-ray, time-series, tabular, Inertial Measurement Unit (IMU), and graph data. It reveals the potential of transformer architectures for unified multi-modal intelligence.

## ABSTRACT

Multimodal learning aims to build models that can process and relate information from multiple modalities. Despite years of development in this field, it still remains challenging to design a unified network for processing various modalities (*e.g.* natural language, 2D images, 3D point clouds, audio, video, time series, tabular data) due to the inherent gaps among them. In this work, we propose a framework, named Meta-Transformer, that leverages a **frozen** encoder to perform multimodal perception without any paired multimodal training data. In Meta-Transformer, the raw input data from various modalities are mapped into a shared token space, allowing a subsequent encoder with frozen parameters to extract high-level semantic features of the input data. Composed of three main components: a unified data tokenizer, a modality-shared encoder, and task-specific heads for downstream tasks, Meta-Transformer is the first framework to perform unified learning across 12 modalities with unpaired data. Experiments on different benchmarks reveal that Meta-Transformer can handle a wide range of tasks including fundamental perception (text, image, point cloud, audio, video), practical application (X-Ray, infrared, hyperspectral, and IMU), and data mining (graph, tabular, and time-series). Meanwhile, it also excels in multimodal understanding on cross-modal retrieval, referring segmentation, and grounding tasks. Meta-Transformer indicates a promising future for developing unified multimodal intelligence with transformers. We will release well-documented code and pretrained weights soon.

# 1 INTRODUCTION

The human brain, which is considered the inspiration for neural network models, processes information from various sensory inputs, *e.g.* visual, auditory, and tactile signals, simultaneously. Moreover, the brain simultaneously learns multi-sensory knowledge efficiently. However, in deep learning, it is significantly invaluable and meaningful to design a unified network capable of processing a wide range of data formats of high efficiency due to challenging modality gaps (Wang et al., 2021d;c; 2022c).

Each data modality presents unique data patterns, which makes it difficult to adapt models trained on one modality to another. For instance, images exhibit a high degree of information redundancy due to densely packed pixels, which is not the case with natural language (He et al., 2022). Point clouds, on the other hand, have a sparse distribution in 3D space, making them more susceptible to noise and challenging to represent (Qi et al., 2017a). Audio spectrograms are time-varying and non-stationary data patterns consisting of combinations of waves across frequency domains (Gong et al., 2021). Video data contains a sequence of image frames, which gives it the unique capability to capture both spatial information and temporal dynamics (Bertasius et al., 2021). Graph data represents entities as nodes and relationships as edges in a graph, modeling complex, many-to-many relationships between entities (Gilmer et al., 2017). Owing to the substantial differences inherent to various data modalities, it is common practice to utilize distinct network architectures to encode each modality separately. For instance, Point Transformer (Zhao et al., 2021) leverages vector-level position attention to extract structural information from 3D coordinates, but it cannot encode an image, a natural language paragraph, or an audio spectrogram slice. Therefore, designing a unified framework capable of utilizing a modality-shared parameter space to encode multiple data modalities remains a significant challenge. Recently, the development of unified frameworks such as VLMO (Wang et al., 2021c), OFA (Wang et al., 2022a), and BEiT-3 (Wang et al., 2022c) have improved the ability of the network for multimodal understanding, through large-scale multimodal pretraining on paired data (Wang et al., 2022c;a; 2021c), but they are more focused on vision and language, and unable to share the whole encoder across modalities.

The transformer architecture and attention mechanism, proposed by Vaswani et al. (2017) for natural language processing (NLP), have made a significant difference in deep learning (Vaswani et al., 2017; Carion et al., 2020b; Dosovitskiy et al., 2021a; Zhai et al., 2022; Xie et al., 2021; Wang et al., 2021a). These advancements have been instrumental in enhancing perception across different modalities such as 2D vision (Dosovitskiy et al., 2021b; Chen et al., 2022; Liu et al., 2021b), 3D vision (Zhao et al., 2021; Yu et al., 2022; Qian et al., 2022b), audio signal processing (Gong et al., 2021) , *etc*. These works have demonstrated the versatility of transformer-based architectures, inspiring researchers to explore *whether it's possible to develop foundation models capable of unifying multiple modalities, ultimately achieving human-level perception across all modalities.*

Table 1: Comparison between Meta-Transformer and related works on perception tasks.

| Method | Modalities | Share Parameters | Unpaired Data |
|---|---|---|---|
| Transformer | 📄 | ✗ | ✗ |
| ViT, Swin Transformer, MAE | 🌎 | ✗ | ✗ |
| Point Transformer, PCT, Point ViT | 🏹 | ✗ | ✗ |
| AST, SSAST | 📠 | ✗ | ✗ |
| CLIP, Flamingo, VLMO, OFA | 📄🌎 | ✗ | ✗ |
| BEiT-3 | 📄🌎 | Several Layers | ✗ |
| ImageBind | 📄🌎🏹📠📺🔥🎿 | ✗ | ✗ |
| Meta-Transformer [ours] | 📄🌎🏹📠📺⬜🎿🔥🎽🔥 | Whole Backbone | ✔ |

In this paper, we explore the potential of transformer architecture to process 12 modalities including images, natural language, point cloud, audio spectrogram, video, infrared, hyperspectral, X-Ray, IMU, tabular, graph, and time-series data, as shown in Figure 1. We discuss the learning process with transformers for each modality and address the challenges associated with unifying them into a single framework. Consequently, we propose a novel unified framework named Meta-Transformer for multimodal learning. **Meta-Transformer is the first framework to simultaneously encode data from a dozen of modalities using the same set of parameters**, allowing a more cohesive approach to multimodal learning (as shown in Table 1). Meta-Transformer incorporates three simple and effective components: a modality-specialist (§ 3.2) for data-to-sequence tokenization, a modality-shared encoder (§ 3.3) for extracting representations across modalities, and task-specific heads

for downstream tasks. Specifically, Meta-Transformer first utilizes modality-specific tokenizers to transform multimodal data into token sequences that share a common manifold space. Then, a modality-shared encoder with frozen parameters is used to extract representations. Finally, the representations will be input into different downstream task heads. With this simple framework, both task-specific and modality-generic representations can be effectively learned, from unpaired data.

We conduct extensive experiments on various benchmarks of 12 modalities. By utilizing images of LAION-2B (Radford et al., 2021) dataset for pretraining exclusively, Meta-Transformer demonstrates remarkable performance in processing data from multiple modalities, consistently achieving superior performances over state-of-the-art methodologies across different multimodal learning tasks. More detailed experimental settings can be found in § D.

In conclusion, our contributions can be summarized as follows:

- For multimodal research, we propose a novel framework, Meta-Transformer, which utilizes a unified encoder to simultaneously extract representations from multiple modalities with the same set of parameters.

- For multimodal network design, we comprehensively examine the functions of transformer components (*e.g.* embeddings, tokenization) and encoders in processing various modalities. Meta-Transformer provides valuable insights and sparks a promising new direction in developing a modality-agnostic foundation model capable of unifying all modalities.

- Experimentally, Meta-Transformer achieves outstanding performance on various datasets spanning 12 modalities and excels in multimodal understanding, which validates the further potential of Meta-Transformer for unified multimodal learning.

## 2 RELATED WORK

### 2.1 SINGLE-MODALITY PERCEPTION

**Multi-Layer Perceptron for pattern recognition.** At the beginning, support vector machine (SVM) and multi-layer perceptron (MLP) are applied to text (Xu et al., 2003), image (LeCun et al., 1989), point cloud (Qi et al., 2017b), and audio (Dhanalakshmi et al., 2009) classification.

**Recurrent & Convolutional Neural Network.** Hopfield Network (Hopfield, 1982) is the original form of recurrent networks, then LSTM (Hochreiter & Schmidhuber, 1997) and GRU (Chung et al., 2014) further explore the advantages of RNNs in sequence modeling and application in NLP tasks (Nallapati et al., 2016; Cho et al., 2014; Tang et al., 2015), which is also widely applied in audio synthesis (Kalchbrenner et al., 2018). Meanwhile, the success of CNNs including LeNet (LeCun et al., 1998), AlexNet (Krizhevsky et al., 2017), VGG (Simonyan & Zisserman, 2015), GoogleNet (Szegedy et al., 2015) and ResNet (He et al., 2016) in image recognition greatly promote the application of CNNs in other fields such as text classification (Zhang et al., 2015; Zhang & Wallace, 2015), point cloud understanding (Li et al., 2018; Maturana & Scherer, 2015; Thomas et al., 2019), and speech classification (Abdel-Hamid et al., 2014).

**Transformer.** Recently, transformer architecture (Vaswani et al., 2017) has been adopted in various tasks such as text understanding (Devlin et al., 2019) and generation (Brown et al., 2020) in NLP, classification (Dosovitskiy et al., 2021a), detection (Carion et al., 2020a) and segmentation (Xie et al., 2021) in images, point cloud understanding (Guo et al., 2021; Zhao et al., 2021), and audio recognition (Gong et al., 2021; 2022).

### 2.2 TRANSFORMED-BASED MULTIMODAL PERCEPTION

Yu et al. (2019) proposes the deep modular co-attention networks between vision and language, which performs the cross-modal alignment. Then it becomes a consensus (Wang et al., 2021c;d; 2022a;c) to utilize a cross-attention mechanism to bridge different modalities. More works are focused on how to effectively align representations extracted across modalities by pretraining. VL-BERT (Su et al., 2019) pioneers modality-aligned representations for generic vision-language understanding. Then Oscar (Li et al., 2020) described the object semantics in both visual and textural contents. Frameworks such as Vinvl (Zhang et al., 2021), Simvlm (Wang et al., 2021d), VLMO (Wang et al.,

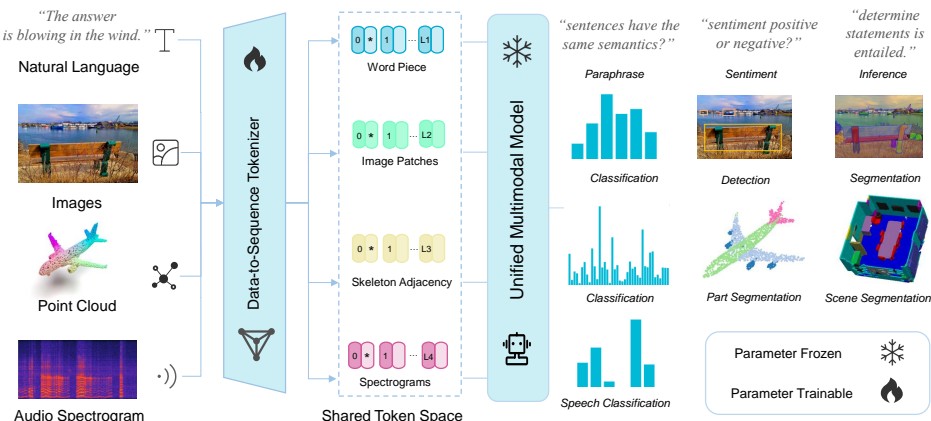

Figure 2: Meta-Transformer consists of data-to-sequence tokenization, unified feature encoding, and down-stream task learning. The framework is illustrated with text, image, point cloud, and audio.

2021c), ALBEF (Li et al., 2021), Florence (Yuan et al., 2021) and Unified-IO (Lu et al., 2022) further explore the advantages of joint representations between text and image. Omnivore (Girdhar et al., 2022) is only focused on visual modalities.

## 3 META-TRANSFORMER

Despite the advances mentioned above, designing unified multimodal networks remains challenging due to the inherent disparities between modalities. Moreover, most research in this area has primarily focused on vision and language tasks, and may not directly contribute to tasks associated with other modalities, such as 3D point cloud understanding, audio recognition, and time-series analysis.

### 3.1 PRELIMINARY

Formally, we denote the input space of $n$ modalities as $\{\mathcal{X}_1, \mathcal{X}_2, \cdots, \mathcal{X}_n\}$, while $\{\mathcal{Y}_1, \mathcal{Y}_2, \cdots, \mathcal{Y}_n\}$ are the corresponding label spaces. In addition, we assume there exists an **effective** parameter space $\Theta_i$ for each modality, where any parameter $\theta_i \in \Theta_i$ can be utilized for processing data $\boldsymbol{x}_i \in \mathcal{X}_i$ from that modality. We say that the essence of Meta-Transformer is to find a shared $\theta^*$ that satisfies:

$$\theta^* \in \Theta_1 \cap \Theta_2 \cap \Theta_3 \cap \cdots \cap \Theta_n, \tag{1}$$

with the hypothesis:

$$\Theta_1 \cap \Theta_2 \cap \Theta_3 \cap \cdots \cap \Theta_n \neq \varnothing. \tag{2}$$

The multimodal neural networks can be formulated as a unified mapping function $\mathcal{F} : \boldsymbol{x} \in \mathcal{X} \to \hat{y} \in \mathcal{Y}$, where $\boldsymbol{x}$ is the input data coming from any modality $\{\mathcal{X}_1, \mathcal{X}_2, \cdots, \mathcal{X}_n\}$ and $\hat{y}$ denotes the prediction of the network. Denoted by $y$ the ground truth labels and $\mathcal{L}$ the loss function, the multimodal pipeline can be formulated as:

$$\hat{y} = \mathcal{F}(\boldsymbol{x}; \theta^*), \ \ \theta^* = \arg\min_{x \in \mathcal{X}} [\mathcal{L}(\hat{y}, y)]. \tag{3}$$

### 3.2 DATA-TO-SEQUENCE TOKENIZATION

We take text, image, point cloud, and audio as examples shown in Figure 3. More details can be found in B.1 and B.3. In specific, we use $\boldsymbol{x}_T$, $\boldsymbol{x}_I$, $\boldsymbol{x}_P$, and $\boldsymbol{x}_A$ to denote a data sample of text, image, point cloud, and audio spectrogram.

**Natural Language**. Following the common practice (Devlin et al., 2019; Liu et al., 2019), we use WordPiece embeddings (Wu et al., 2016) with a 30,000 token vocabulary. WordPiece segments original words into subwords. For example, the original sentence: "The supermarket is hosting a sale", could be converted to: "_The _super market _is _host ing _a _sale". Each subword corresponds

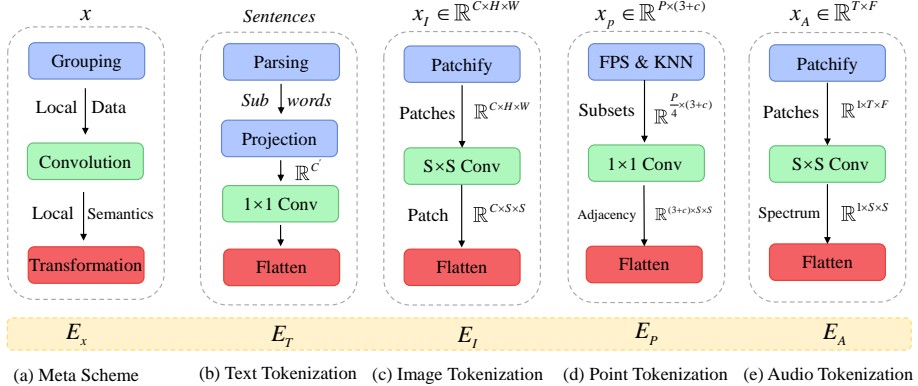

Figure 3: Illustration of Data-to-Sequence Tokenization 3.2. We propose the meta scheme in (a) containing grouping, convolution, and transformation progress. Then (b)-(e) represents the building blocks applied with our meta scheme on texts, images, point clouds, and audio spectrograms.

to a unique token in a vocabulary, and then gets projected to a high-dimensional feature space with word embedding layers. As a result, each input text is transformed to a set of token embeddings $\boldsymbol{x} \in \mathbb{R}^{n \times D}$, where $n$ is the number of tokens and $D$ is the dimension of embedding.

**Image**. To accommodate 2D images, we reshape the image $\boldsymbol{x} \in \mathbb{R}^{H \times W \times C}$ into a sequence of flattened 2D patches $\boldsymbol{x}_p \in \mathbb{R}^{N_s \times (S^2 \cdot C)}$, where $(H, W)$ represents the original image resolution, $C$ denotes the number of channels; $S$ is the patch size, and $N_s = (HW/S^2)$ is the resulting number of patches. After that, a projection layer is utilized to project the embedding dimension to $D$:

$$\boldsymbol{x}_I \in \mathbb{R}^{C \times H \times W} \to \boldsymbol{x}'_I \in \mathbb{R}^{N_s \times (S^2 \cdot C)} \to \boldsymbol{x}''_I \in \mathbb{R}^{N_s \times D}. \tag{4}$$

**Point Cloud**. To learn 3D patterns with transformers, we convert point clouds from raw input space to the token embedding space. $\mathcal{X} = \{\boldsymbol{x}_i\}_{i=1}^P$ denotes a point cloud of $P$ points, where $\boldsymbol{x}_i = (\boldsymbol{p}_i, \boldsymbol{f}_i)$, $\boldsymbol{p}_i \in \mathbb{R}^3$ represents the 3D coordinates, and $\boldsymbol{f}_i \in \mathbb{R}^c$ is feature of the $i$-th point. Generally, $\boldsymbol{f}_i$ contains visual hints such as color, viewpoint, normal, etc. We employ the Farthest Point Sampling (FPS) operation to sample a representative skeleton of original point clouds. Then we employ $K$-Nearest Neighbor (KNN) to group neighboring points. We construct the adjacency matrix with center points of grouped subsets. Finally, we aggregate the structural representations from $K$ subsets. We obtain point embeddings as:

$$\boldsymbol{x}_P \in \mathbb{R}^{P \times (3+c)} \to \boldsymbol{x}'_P \in \mathbb{R}^{\frac{P}{4} \times \frac{D}{2}} \to \boldsymbol{x}''_P \in \mathbb{R}^{\frac{P}{16} \times D}. \tag{5}$$

**Audio Spectrogram**. Initially, we pre-process the audio waveform with the duration of $t$ seconds with log Mel filterbank (Schneider et al., 2019). Then we employ the Hamming window with a stride of $t_s$ on the frequency of $f_s$ to split the original wave into $l = (t/t_s)$ intervals and further transform the original wave into $l$-dimensional filterbank. Subsequently, we split the spectrogram into patches from time and frequency dimensions with the same patch size of $S$. Different from image patches, audio patches overlap on spectrograms. Following AST (Gong et al., 2021), we choose to split the whole spectrograms into $N_s = 12[(100t - 16)/10]$ patches by $S \times S$ convolution, then we flatten patches into token sequences. The whole process can be summarized as:

$$\boldsymbol{x}_A \in \mathbb{R}^{T \times F} \to \boldsymbol{x}'_A \in \mathbb{R}^{N_s \times S \times S} \to \boldsymbol{x}''_A \in \mathbb{R}^{(N_s \cdot D/S^2) \times D}, \tag{6}$$

where $T$ and $F$ denote time and frequency dimension.

## 3.3 UNIFIED MULTIMODAL ENCODER

After transforming the raw input into token embedding space, we leverage a unified transformer encoder with frozen parameters to encode the token embedding sequences from different modalities.

**Pretraining**. We utilize ViT (Dosovitskiy et al., 2021a) as the backbone network and pre-train it on the LAION-2B dataset with contrastive learning, which reinforces the ability for generic token

encoding. After pretraining, we freeze the parameters of the backbone network. In addition, for text understanding, we utilize the pretrained text tokenizer of CLIP (Radford et al., 2021) to segment sentences into subwords and transform subwords into word embeddings.

**Modality-Agnostic Learning**. Following common practice (Devlin et al., 2019; Dosovitskiy et al., 2021a), we prepend a learnable token $x_{\text{CLS}}$ to the sequence of token embeddings, and the final hidden state of $x_{\text{CLS}}$ token ($z_L^0$) serves as the summary representation of the input sequence, which is usually utilized for performing recognition. The transformer encoder with a depth of $L$ comprises multiple stacked multi-head self-attention (MSA) layers and MLP blocks. The input token embeddings are fed into an MSA layer first, and then an MLP block. The output of $(\ell - 1)$-th MLP block serves as the input of $\ell$-th MSA layer. Layer Normalization (LN) is appended before each layer and the residual connection is applied after each layer. The MLP contains two linear FC layers along with a GELU non-linear activation. Thus transformer can be formulated as the following:

$$z_0 = [x_{\text{CLS}}; E_{x_1}; E_{x_2}; \cdots; E_{x_n}] + E_{pos}, \qquad E \in \mathbb{R}^{n \times D}, \ E_{pos} \in \mathbb{R}^{(n+1) \times D} \qquad (7)$$

$$z'_\ell = \text{MSA}(\text{LN}(z_{\ell-1})) + z_{\ell-1}, \qquad \ell = 1 \ldots L \qquad (8)$$

$$z_\ell = \text{MLP}(\text{LN}(z'_\ell)) + z'_\ell, \qquad \ell = 1 \ldots L \qquad (9)$$

$$y = \text{LN}(z_L^0), \qquad (10)$$

where $E_x$ denotes the token embeddings from proposed tokenizer and $n$ denotes the number of tokens. We augment patch embeddings and learnable embedding with position embeddings $E_{pos}$.

### 3.4 TASK-SPECIFIC HEADS

After the unified feature encoder, the obtained representations are input into the task-specific heads $h(\cdot; \theta_h)$, which consist mainly of MLPs and vary across modalities and tasks. The overall learning objective of Meta-Transformer can be summarized as:

$$\hat{y} = \mathcal{F}(x; \theta^*) = h \circ g \circ f(x), \quad \theta^* = \arg\min_\theta \mathcal{L}(\hat{y}, y), \qquad (11)$$

where $f(\cdot)$, $g(\cdot)$, and $h(\cdot)$ denote the function of tokenizer, backbone, and task heads, respectively.

## 4 EXPERIMENTS

In this section, we perform experiments on each of the 12 modalities (§ 4.1), and we demonstrate Table 2: **Single-Modality Perception**. Summary of experimental settings across different modalities. We report the task, dataset, data scale, loss function, task head, and the ratio of trainable parameters for each modality.

| Modalities | Tasks | Datasets | Data Scale | Loss Function | Head | Ratio |
|---|---|---|---|---|---|---|
| Text | Classification | GLUE Benchmark | 330K | Cross Entropy | Linear Layers | <1% |
| Image | Classification | ImageNet-1K | 1.3M | Smooth Cross Entropy | Linear Layers | <1% |
| | Detection | MS COCO | 118K | Focal & IoU Loss | Mask RCNN | 39.8% |
| | Segmentation | ADE-20K | 20K | Cross Entropy | UpperNet | 47.6% |
| Point Cloud | Shape Classification | ModelNet-40 | 9K | Smoth Cross Entropy | Linear Layers | <1% |
| | Scene Segmentation | S3DIS | 400M Points | Cross Entropy | Convolution Layers | 2.6% |
| | Object Segmentation | ShapeNetPart | 16K | Poly1 FocalLoss | Convolution Layers | 2.6% |
| Audio | Classification | Speech commands v2 | 105K | Cross Entropy | Linear Layers | 1.3% |
| Video | Action Recognition | UCF101 | 14K | Soft Cross Entropy | Linear Layers | 1.3% |
| Infrared | Classification | RegDB | 40K | Cross Entropy & Center & Triplet Loss | Linear Layers | <1% |
| Hyper-spectrum | Classification | Indian Pine | 10K | Cross Entropy | Linear Layers | <1% |
| X-Ray | Classification | Chest X-Ray | 112K | Cross Entropy | Linear Layers | <1% |
| IMU | Classification | Ego4D | 193K | Cross Entropy | Linear Layers | <1% |
| Tabular data | Prediction | Adult & Bank | 32K-45K | Binary Cross Entropy | Linear Layers | <1% |
| Graph data | Prediction | PCQM4M-LSC | 47M | L1 Loss | Linear Layers | <1% |
| Time-series | Forecasting | Exchange, Traffic, *etc* | 5-36K | MSE Loss | Transformer Decoder | 8.5% |

the potential of Meta-Transformer for multimodal perception (§ 4.2). Following ViT (Dosovitskiy et al., 2021a), Meta-Transformer-B16$_F$ denotes a base-scale encoder which contains 12 transformer blocks and 12 attention heads, and the image patch size is 16. And the embedding dimension is 768, the output dimension of MLP is 3,072. 'F' and 'T' denote that parameters of the encoder are *Frozen* and further *Tuned*, respectively.

### 4.1 SINGLE-MODALITY PERCEPTION

we summarize the evaluation experiments as shown in Table 2, more details can be found in Appendix D.

Table 3: **Experimental results for text understanding on the GLUE benchmark.** We compare existing methods from paraphrasing, sentiment, duplication, inference, and answering tasks.

| Method | Pretraining Settings | | | GLUE Benchmark | | | | |
|---|---|---|---|---|---|---|---|---|
| | Modality | Data | Size | SST-2 Sentiment | MRPC Paraphrase | QQP Duplication | MNLI Inference | QNLI Answering |
| BiLSTM+ELMo+Attn | - | - | - | 90.4 | 84.9 | 64.8 | 76.4 | 79.8 |
| OpenAI GPT (Radford et al., 2018) | | Book | 0.8B | 91.3 | 82.3 | 70.3 | 82.1 | 87.4 |
| BERT$_{BASE}$ (Devlin et al., 2019) | Language | Wiki+Book | 3.3B | 88.0 | 88.9 | 71.2 | 84.6 | 90.5 |
| RoBERTa$_{BASE}$ (Liu et al., 2019) | | | | **96.0** | **90.0** | **84.0** | 84.0 | **92.0** |
| ChatGPT | | Various | 4,5000B | 92.0 | 66.0 | 78.0 | **89.3** | 84.0 |
| Meta-Transformer-B16$_F$ [ours] | Image | LAION-2B (Radford et al., 2021) | 2B | 54.6 | 81.1 | 66.0 | 63.4 | 56.3 |
| Meta-Transformer-B16$_T$ [ours] | | | | 81.3 | 81.8 | 78.0 | 70.0 | 60.3 |

Table 4: **Experimental results for image understanding**. We conduct experiments on the ImageNet (Deng et al., 2009), MSCOCO (Lin et al., 2014), and ADE-20K (Zhou et al., 2017) datasets, where **Bold** and underline indicate best and second best results.

| Method | Classification | | | | Object Detection | | | Semantic Segmentation | | |
|---|---|---|---|---|---|---|---|---|---|---|
| | Res | #Params | #FLOPs | Acc (%) | #Params | #FLOPs | AP (%) | #Params | #FLOPs | mIoU (%) |
| PVT-L (Wang et al., 2021b) | $224^2$ | 61.4M | 9.8G | 81.7 | 81.0M | - | 42.9 | 65.1M | 79.6G | 44.8 |
| Swin-L$^\ddagger$ (Liu et al., 2021b) | $384^2$ | 197M | 104G | 87.3 | 253M | 1382G | 51.8 | 234M | 2468G | 52.1 |
| CoAtNet-4$^\ddagger$ (Dai et al., 2021) | $384^2$ | 275M | 190G | 87.9 | - | - | - | - | - | - |
| DeiT III-L$^\ddagger$ (Touvron et al., 2022) | $384^2$ | 304M | 191G | 87.7 | - | - | - | 353.6M | 2231G | 51.5 |
| SwinV2-L/24$^\ddagger$ (Liu et al., 2022b) | $384^2$ | 197M | 115G | 87.6 | - | - | **58.8** | - | - | **55.9** |
| RepLKNet-31L$^\ddagger$ (Ding et al., 2022) | $384^2$ | 172M | 96G | 86.6 | 229M | 1321G | 53.9 | 207M | 2404G | 52.4 |
| HorNet-L$^\ddagger$ (Rao et al., 2022) | $384^2$ | 202M | 102G | 87.7 | 259M | 1358G | 56.0 | 232M | 2473G | 54.1 |
| ConvNeXt-L$^\ddagger$ (Liu et al., 2022d) | $384^2$ | 198M | 101G | 87.5 | 255M | 1354G | 53.5 | 235M | 2458G | 53.2 |
| InternImage-L$^\ddagger$ (Wang et al., 2022b) | $384^2$ | 223M | 108G | 87.7 | 277M | 1399G | 54.9 | 256M | 2526G | 53.9 |
| InternImage-XL$^\ddagger$ (Wang et al., 2022b) | $384^2$ | 335M | 163G | 88.0 | 387M | 1782G | 55.3 | 368M | 3142G | 55.0 |
| Meta-Transformer-B16$_F$ [ours] | $224^2$ | 86.6M | 17.5G | 69.3* | 143M | 1126G | 31.7 | 164M | 135G | 33.4 |
| | $224^2$ | 86.6M | 17.5G | 79.3$^\dagger$ | | | | | | |
| Meta-Transformer-L14$_F$ [ours] | $336^2$ | 191.1M | 190.6G | 75.3* | 364M | 2143G | 43.5 | 314M | 683G | 41.2 |
| | $336^2$ | 191.1M | 190.6G | 83.1$^\dagger$ | | | | | | |
| Meta-Transformer-B16$_T$ [ours] | $224^2$ | 86.6M | 17.5G | 85.4 | 143M | 1126G | 46.4 | 164M | 135G | 48.3 |
| Meta-Transformer-L14$_T$ [ours] | $336^2$ | 191.1M | 190.6G | **88.1** | 364M | 2143G | 56.3 | 314M | 683G | 55.0 |

*: zero-shot classification    $^\dagger$: linear probing for classification    $^\ddagger$: models pre-trained on ImageNet-22K

**Results on Natural Language Understanding** Table 3 illustrates the experimental results on the GLUE benchmark for text understanding tasks, Meta-Transformer-B16$_T$ exhibits improved performance, with 81.3% in sentiment, 81.8% in paraphrase, 78.0% in duplication, 70.0% in inference, and 60.3% in answering tasks.

**Results on Image Understanding** As shown in Table 4, Meta-Transformer exhibits outstanding performance on image understanding tasks. It delivers great performances in classification with Meta-Transformer-B16$_T$ and Meta-Transformer-L14$_T$ achieving 85.4% and 88.1% accuracy, respectively. When it comes to object detection and semantic segmentation, Meta-Transformer-L14$_T$ has a similar performance to InternImage-XL$^\ddagger$ (Wang et al., 2022b) in semantic segmentation, but outperforms it in object detection.

**Results on Infrared, Hyperspectral, and X-Ray data**. Table 5a shows that Meta-Transformer-B16$_F$ delivers competitive results with a Rank-1 accuracy of 73.50% and an mAP of 65.19%.

In addition, Table 5b presents the performance of Meta-Transformer on the Indian Pine dataset for hyperspectral image recognition. Meta-Transformer stands out for its significantly fewer trainable parameters (only 0.17M) compared to other methods. This reveals a promising development direction of applying the Meta-Transformer to remote sensing, environmental monitoring, and mineral exploration. For X-Ray images, in Table 9, we can observe that Meta-Transformer can achieve a competitive performance of 94.1% accuracy.

Table 5: **Experimental results for infrared and hyperspectral data understanding**. We conduct experiments on classification tasks over the RegDB and Indian Pine datasets. We report Rank-1 (R@1), mean Average Precision (mAP), Overall Accuracy (OA), Average Accuracy (AA), and the number of trainable parameters (Params).

| Method | R@1 (%) | mAP (%) | Params |
|---|---|---|---|
| AGW (Ye et al., 2020) [TPAMI'21] | 70.49 | 65.90 | 25M |
| SMCL (Wei et al., 2021) [ICCV'21] | 83.05 | **78.57** | 40M |
| MSCLNet (Zhang et al., 2022) [ECCV'22] | 83.86 | 78.31 | 50M |
| Meta-Transformer-B16$_F$ | 73.50 | 65.19 | **1.8M** |

| Method | OA (%) | AA (%) | Params |
|---|---|---|---|
| ViT (Dosovitskiy et al., 2021a) [ICLR'21] | 71.86 | 78.97 | 85.2M |
| SpectralFormer (Hong et al., 2021) [TGRS'21] (Pixel) | 78.55 | 84.68 | 85.2M |
| SpectralFormer (Hong et al., 2021) [TGRS'21] (Patch) | 81.76 | 87.81 | 85.2M |
| Meta-Transformer-B16$_F$ | 67.62 | 78.09 | **0.17M** |

(a) Infrared data understading        (b) Hyperspectral data understanding

**Results on 3D Point Cloud Understanding** Table 6 showcases the experimental results for point cloud understanding, comparing the performance of Meta-Transformer with other state-of-the-art methods on the ModelNet-40 (Wu et al., 2015), S3DIS (Armeni et al., 2016), and ShapeNetPart (Yi et al., 2016) datasets.

Table 6: **Experimental results for point cloud understanding**. We conduct experiments on the ModelNet-40 (Wu et al., 2015), S3DIS (Armeni et al., 2016), and ShapeNetPart (Yi et al., 2016) datasets.

| Method | Pre-train | ModelNet-40 | | | S3DIS Area-5 | | | ShapeNetPart | | |
|---|---|---|---|---|---|---|---|---|---|---|
| | | mAcc (%) | OA (%) | Params | mIoU (%) | mAcc (%) | Params | mIoU_I (%) | mIoU_C (%) | Params |
| PointNet [CVPR'17] (Qi et al., 2017b) | N/A | 86.0 | 89.2 | 3.5M | 41.1 | 49.0 | 3.6M | 83.7 | 80.4 | 3.6M |
| PointNet++ [NeurIPS'17] (Qi et al., 2017a) | N/A | - | 91.9 | 1.5M | 53.5 | - | 1.0M | 85.1 | 81.9 | 1.0 |
| PointCNN [NeurIPS'18] (Li et al., 2018) | N/A | 88.1 | 92.5 | 0.6M | 57.3 | - | 0.6M | | | |
| DGCNN [TOG'19] (Wang et al., 2019) | N/A | 90.2 | 92.9 | 1.8M | 52.5 | - | 1.3M | 85.2 | 82.3 | 1.3 |
| Point Transformer [ICCV'21] (Zhao et al., 2021) | N/A | 90.6 | 93.7 | 7.8M | 70.4 | - | 7.8M | 86.6 | 83.7 | 7.8 |
| PointNeXt [NeurIPS'22](Qian et al., 2022a) | N/A | 90.8 | 93.2 | 1.4M | 67.3 | 73.9 | 3.8M | 86.7 | 84.4 | 1.0 |
| Point-MLP [ICLR'22] (Ma et al., 2022) | N/A | 90.9 | 93.6 | 0.68M | - | - | - | 86.1 | 84.6 | - |
| PointMixer [ECCV'22] (Choe et al., 2022) | N/A | 91.4 | 93.6 | 3.6M | 71.4 | 77.4 | 6.5M | - | - | - |
| Point-BERT [CVPR'22] (Yu et al., 2022) | 3D | - | 93.2 | 21.1M | 60.8 | 69.9 | 21.1M | 85.6 | 84.1 | 21.1M |
| Point-MAE [ECCV'22] (Pang et al., 2022) | 3D | - | **93.8** | 21.1M | - | - | - | 86.1 | 84.2 | 21.1M |
| P2P [NeurIPS'22] (Wang et al., 2022d) | 2D | - | 93.1 | 1.2M | - | - | - | 86.5 | 84.1 | - |
| Meta-Transformer-B16F [ours] | 2D | 90.5 | 93.6 | 0.6M | **72.3** | **83.5** | 2.3M | **87.0** | **85.2** | 2.3M |

Table 7: **Audio understanding with Meta-Transformer**. We conduct experiments on the Speech Commands V2 dataset and report the accuracy and numbers of trainable and all parameters.

| Method | Pre-train | Acc (%) | A-Params | Params |
|---|---|---|---|---|
| AST (Gong et al., 2021) (Supervised) | N/A | 92.6 | 86.9M | 86.9M |
| AST (Gong et al., 2021) (Supervised) | AudioSet-20K | 96.2 | 86.9M | 86.9M |
| AST (Gong et al., 2021) (Supervised) | ImageNet+KD | **98.1** | 86.9M | 86.9M |
| SSAST (Gong et al., 2022) (Self-Supervised) | AudioSet-2M | 97.8 | 89.3M | 89.3M |
| SSAST (Gong et al., 2022) (Self-Supervised) | Librispeech | 97.8 | 89.3M | 89.3M |
| SSAST (Gong et al., 2022) (Self-Supervised) | Joint Pretraining | 98.0 | 89.3M | 89.3M |
| Meta-Transformer-B16F [ours] | 2D | 78.3 | 86.6M | **1.1M** |
| Meta-Transformer-B16T [ours] | 2D | 97.0 | 86.6M | 86.3M |

Meta-Transformer demonstrates remarkable advantages in point cloud understanding tasks, offering competitive performance with fewer trainable parameters compared to other state-of-the-art methods.

**Results on Audio Recognition**
Table 7 shows the performance of Meta-Transformer in the audio understanding. Compared to AST (Gong et al., 2021) and SSAST (Gong et al., 2022) on accuracy, with frozen parameters, Meta-Transformer-B16F achieves an accuracy of 78.3%.

**Results on Video Recognition** Table 8a presents the performance comparison of the Meta-Transformer and existing advanced methods on the UCF101 dataset for video understanding. Meta-Transformer stands out for its significantly reduced trainable parameter count, suggesting the potential benefit of unified multi-modal learning and less architectural complexity.

Table 8: **Experimental results for video and tabular data understanding**.

| Method | Modality | UCF101 | Params |
|---|---|---|---|
| OPN (Lee et al., 2017) | V | 59.6 | - |
| SimCLR (Feichtenhofer et al., 2021) | V | 88.9 | 86.9M |
| VideoMAE V1 (Tong et al., 2022) | V | 96.1 | 86.9M |
| VideoMAE V2 (Wang et al., 2023) | V | **99.6** | 86.9M |
| ViT (Dosovitskiy et al., 2021a) (from scratch) | V | 51.4 | 86.9M |
| Meta-Transformer-B16F | V | 46.6 | **1.1M** |

| Method | Adult Accuracy (%) | Bank Marketing Accuracy (%) | F1 |
|---|---|---|---|
| LightGBM | 87.8 | - | 0.39 |
| Tabmlp | 87.2 | - | 0.39 |
| Tabnet | 87.0 | - | 0.31 |
| Tabtransformer | 87.1 | 93.4 | 0.42 |
| Meta-Transformer-B16F | 85.9 | 90.1 | 0.41 |

(a) Video understanding     (b) Tabular data understanding

Table 9: X-ray recognition on Chest X-Ray dataset.

| Method | Accuracy (%) | Params |
|---|---|---|
| ViT (Dosovitskiy et al., 2021a) | 96.3 | 86.9M |
| SEViT (Almalik et al., 2022) | 94.6 | 85.8M |
| Meta-Transformer-B16F | 94.1 | **0.75M** |

**Results on Time-series Forecasting**
From Table 10, 1) with most of the model parameters being fixed, our method can still outperform existing methods including Pyraformer (Liu et al., 2021a), Informer (Zhou et al., 2021), LogTrans (Li et al., 2019), and Reformer (Kitaev et al., 2020). 2) With only 19K trainable parameters, Meta-Transformer can still outperform Informer (Zhou et al., 2021). Therefore, Meta-Transformers pretrained on perception tasks can also be applied to time-series forecasting tasks, which is inspiring for this area.

Table 10: **Time-series Forecasting with Meta-Transformer**. Following TimesNet, we report the number of trainable parameters and average performances from 4 different prediction lengths, which is {96, 192, 336, 720}.

| Models | Meta-Transformer [Ours] | | | TimesNet | | ETSformer | | FEDformer | | Stationary | | Autoformer | | Pyraformer | | Informer | | LogTrans | | Reformer | |
|---|---|---|---|---|---|---|---|---|---|---|---|---|---|---|---|---|---|---|---|---|---|
| Metric | MSE | MAE | Param | MSE | MAE | MSE | MAE | MSE | MAE | MSE | MAE | MSE | MAE | MSE | MAE | MSE | MAE | MSE | MAE | MSE | MAE |
| ETTh1 | 0.994 | 0.797 | **19K** | 0.458 | **0.450** | 0.542 | 0.510 | **0.440** | 0.460 | 0.570 | 0.537 | 0.496 | 0.487 | 0.827 | 0.703 | 1.040 | 0.795 | 1.072 | 0.837 | 1.029 | 0.805 |
| Traffic | 0.694 | 0.372 | **2.0M** | 0.620 | **0.336** | 0.621 | 0.396 | **0.610** | 0.376 | 0.624 | 0.340 | 0.628 | 0.379 | 0.878 | 0.469 | 0.764 | 0.416 | 0.705 | 0.395 | 0.741 | 0.422 |
| Weather | 0.797 | 0.640 | **51K** | **0.259** | **0.287** | 0.271 | 0.334 | 0.309 | 0.360 | 0.288 | 0.314 | 0.338 | 0.382 | 0.946 | 0.717 | 0.634 | 0.548 | 0.696 | 0.602 | 0.803 | 0.656 |
| Exchange | 1.430 | 0.961 | **22K** | 0.416 | 0.443 | **0.410** | **0.427** | 0.519 | 0.500 | 0.461 | 0.454 | 0.613 | 0.539 | 1.913 | 1.159 | 1.550 | 0.998 | 1.402 | 0.968 | 1.280 | 0.932 |

Table 11: **Graph data understanding with Meta-Transformer**. We conduct experiments on the PCQM4M-LSC dataset.

| Method | Param. | train MAE | validate MAE |
|---|---|---|---|
| GCN | 2.0M | 0.1318 | 0.1691 |
| GIN | 3.8M | 0.1203 | 0.1537 |
| GCN-$_{VN}$ | 4.9M | 0.1225 | 0.1485 |
| GIN-$_{VN}$ | 6.7M | 0.1150 | 0.1395 |
| GINE-$_{VN}$ | 13.2M | 0.1248 | 0.1430 |
| DeeperGCN-$_{VN}$ | 25.5M | 0.1059 | 0.1398 |
| Graph Transformer | 0.6M | 0.0944 | 0.1400 |
| Graph Transformer-$_{Wide}$ | 83.2M | 0.0955 | 0.1408 |
| Graphormer$_{SMALL}$ | 12.5M | 0.0778 | 0.1264 |
| Graphormer | 47.1M | **0.0582** | **0.1234** |
| Meta-Transformer-B16$_F$ | 1.1M | 0.8034 | 0.8863 |

**Results on Tabular Data Understanding**. Table 8b provides the comparison betweendifferent methods for tabular data understanding. Meta-Transformer-B16$_F$ achieves a competitive accuracy on Adult Census but performs better than others on Bank Marketing dataset.

**Results on Graph and IMU Data Understanding**. In Table 11, Meta-Transformer-B16$_F$ delivers the train and validation MAE scores of 0.8034 and 0.8863, which reveals the limited ability for structural data learning. Besides, following ImageBind (Girdhar et al., 2023), we conduct classification on the Ego4D dataset (Grauman et al., 2022), with input data, Meta-Transformer delivers an accuracy of 73.9%.

## 4.2 MULTI-MODALITY PERCEPTION

In addition to single-modality perception tasks, we also evaluate Meta-Transformer on multimodal tasks. Without any specific network design for cross-modal fusion, we simply concatenate multimodal embeddings and feed them to Meta-Transformer. Compared with existing methods, our method delivers outstanding performance on text-image, audio-visual, and text-3D cross-modal benchmarks.

Table 12: **Multimodal Learning with Meta-Transformer**. We conduct experiments on Text-Image, Audio-Image, and Text-3D perception tasks.

| Method | Venue | Modality | Dataset | Performance (%) |
|---|---|---|---|---|
| **Text Retrieval** | | | | |
| CLIP-L14 | ICML' 21 | 📄 & 🌐 | COCO | R@1 58.4 |
| FLIP-L14 | CVPR' 23 | 📄 & 🌐 | COCO | R@1 60.2 |
| Meta-Transformer-L14 | Ours | 📄 & 🌐 | COCO | R@1 **61.9** ↑1.7 |
| **Image Retrieval** | | | | |
| CLIP-L14 | ICML' 21 | 📄 & 🌐 | COCO | R@1 37.8 |
| FLIP-L14 | CVPR' 23 | 📄 & 🌐 | COCO | R@1 44.2 |
| Meta-Transformer-L14 | Ours | 📄 & 🌐 | COCO | R@1 **46.7** ↑2.5 |
| **Referring Segmentation** | | | | |
| AVSS (ResNet-50) | ECCV' 22 | 🎞 & 🌐 | AVSS | mIoU 20.18 |
| AVSS (PVT-V2) | ECCV' 22 | 🎞 & 🌐 | AVSS | mIoU 29.77 |
| Meta Transformer-B16 | Ours | 🎞 & 🌐 | AVSS | mIoU **31.33** ↑1.56 |
| **3D Visual Grounding** | | | | |
| EDA | CVPR' 23 | ✴ & 📄 | ScanRefer | AP@Unique 85.76 |
| Meta Transformer-B16 | Ours | ✴ & 📄 | ScanRefer | AP@Unique **86.46** ↑0.70 |

In Table 12, we compare Meta-Transformer with existing methods on multimodal tasks. **1**) *Less parameters*: with a shared encoder only, for text-image retrieval, Meta-Transformer outperforms FLIP (Li et al., 2023) by +1.7% for text retrieval and +2.5% for image retrieval on the COCO dataset. **2**) *Faster Convergence*: for audio-visual segmentation, with only 4 training epochs, Meta-Transformer could outperform previous best trained with 30 epochs by +1.56% mIoU. **3**) *Better Performance*: for 3D visual grounding, Meta-Transformer also outperforms EDA (Wu et al., 2022b) by +0.7%. Therefore, we think that *Meta-Transformer demonstrates a more efficient and concise framework for multimodal understanding task*.

## 5 CONCLUSION

In the early stages of artificial intelligence development, pioneers introduced the Multi-Layer Perceptron (MLP) to address prediction tasks in machine learning. Later, recurrent and convolutional networks expanded AI capabilities in multimedia data processing, achieving significant success in extracting representations from texts, images, point clouds, and audio. MLPs have since been integrated into deep convolutional networks. In this paper, we explore the potential of plain transformers for unified multimodal learning, highlighting a promising trend toward developing unified multimodal intelligence with a transformer backbone. To some extent, this paper supports the dominant position of transformers in next-generation networks. Importantly, CNNs and MLPs are not left behind. They play essential roles in data tokenization and representation projection. This process exemplifies the law of succession in neural networks and the ongoing evolution of artificial intelligence.

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

# Appendix

## A SUMMARY

This appendix describes more details of the ICLR 2024 submission, titled *Meta-Transformer: A Unified Framework for Multimodal Learning*. The appendix is organized as follows:

- We detail utilizing Meta-Transformer on more modalities. § B.

- Then we further demonstrate the performance and merits of Meta-Transformer in dealing with multi-modal tasks (involving inputs from more than one modality to perform predictions) in § C.

- In addition, we conduct an ablation study and introduce more details of experiments on text, image, point cloud, audio, and other 8 modalities in § D.

- Beside these details, we also discuss the limitations of Meta-Transformer in § E.

- Last but not least, we discuss the impact of Meta-Transformer on the machine learning and computer vision community in § F.

## B EXTENSIBILITY ON SINGLE-MODALITY PERCEPTION

In the main body of this paper, we illustrate that Meta-Transformer can simultaneously uncover the underlying patterns of natural language, 2D images, 3D point clouds, and audio spectrograms with the same network architecture and network parameters. Furthermore, we explore its ability in perceiving other modalities, like video recognition, infrared, X-Ray, and hyperspectral image recognition. In specific, we conduct experiments on UCF101 (Soomro et al., 2012) (**video**), RegDB (Nguyen et al., 2017) (**infrared** images), Chest **X-Ray** (Rahman et al., 2020), and Indian Pine (**hyperspectral** images) datasets.

### B.1 VIDEO RECOGNITION

For video recognition, we follow VideoMAE (Tong et al., 2022) to modify the tokenizer by replacing the 2D embedding layer with a 3D embedding layer to simultaneously encode the spatial-temporal information from input frames. After tokenization, by leveraging the modality-shared encoder and task-specific heads, Meta-Transformer is able to extract high-level semantic features from videos and achieve favorable performance in the action recognition task of the UCF101 dataset.

**Dataset**. The UCF101 (Soomro et al., 2012) dataset is a common-used benchmark dataset for action recognition tasks. It is an extended version of UCF50 and contains 13,320 video clips of 101 categories. These 101 categories can be divided into 5 groups: Body motion, Human-human interactions, Human-object interactions, Playing musical instruments and Sports. All the input frames are with a resolution of 320×240 and a fixed frame rate of 25 FPS, collected from YouTube.

### B.2 INFRARED IMAGE RECOGNITION

Infrared and hyperspectral image recognition poses unique challenges due to their specific characteristics. For infrared images, the Meta-Transformer framework could be adapted to capture thermal information by encoding temperature values alongside visual features, where the tokenizer for infrared images is the same as common RGB images.

**Dataset**. The RegDB (Nguyen et al., 2017) dataset focuses on evaluating the performance of infrared recognition algorithms in unconstrained and realistic scenarios. It includes variations in pose, expression, illumination, and occlusion. We conduct experiments on the RegDB dataset to evaluate the performance of Meta-Transformer on infrared recognition.

### B.3 HYPERSPECTRAL IMAGE RECOGNITION

Similarly, for hyperspectral images, we expect that Meta-Transformer can also handle the high-dimensional spectral information by representing each spectral band in token embeddings. Compared with dealing with RGB images, the only modification is that we employ the new linear projection layer to replace the existing 2D convolution layer.

**Dataset**. The Indian Pine dataset is widely used in remote sensing and hyperspectral image analysis. It consists of $145 \times 145$ pixels with 145 spectral bands, which are captured in Indiana.

### B.4 X-RAY IMAGE RECOGNITION

In addition, we explore the potential of the Meta-Transformer in medical image analysis. We leverage the tokenizer for RGB images here to encode raw medical images. Specifically, we conduct experiments regarding X-ray image analysis on the Chest X-Ray (Rahman et al., 2020) dataset. It is a collection of medical images commonly used for the analysis and diagnosis of various thoracic conditions. It comprises 7,000 X-ray images of the chest. The dataset is annotated with labels indicating the presence or absence of abnormalities such as lung diseases, fractures, and heart conditions.

## C EXTENSIBILITY ON MULTI-MODALITY PERCEPTION

Since the modalities of text, image, point cloud, and audio are all involved in this paper, we did not conduct comprehensive multi-modal experiments as common practice such as Flamingo (Alayrac et al., 2022), OFA (Wang et al., 2022a), or BEiT-3 (Wang et al., 2022c). Instead, we conduct multi-modal experiments on a new and challenging task of Audio-Visual Segmentation (Zhou et al., 2022a), which is mainly focused on building an intelligent listener to align with fundamental visual tasks.

### C.1 AUDIO-VISUAL SEGMENTATION

Audio-visual segmentation (Zhou et al., 2022a) refers to the task of segmenting objects from different audio sources within a referring image. It aims to develop algorithms that analyze both audio and visual signals simultaneously to identify and delineate distinct sources or events. It finds applications in fields like video conferencing, surveillance, multimedia analysis, and augmented reality.

We conduct experiments on the AVSS (Zhou et al., 2022a) dataset, which is recently released in the field of audio-visual research. It provides a comprehensive collection of audio and visual data captured in real-world scenarios. The dataset includes synchronized audio and visual recordings, featuring various events of human actions and natural sounds. In contrast to introducing multi-modal fusion modules as existing methods, Meta-Transformer directly concatenates visual and audio embeddings after Data-to-Sequence tokenization. After extracting representation, we employ a simple global average pooling layer to obtain the final representations of two modalities. Table 13 illustrates

Table 13: **Audio-Visual Segmentation with Meta-Transformer**. We conduct experiments on the AVSS (Zhou et al., 2022a) dataset, we report mIou (%) and F-score.

| Method | mIou (%) | F-score | Params |
|---|---|---|---|
| AVSS (Zhou et al., 2022a) (ResNet-50) | 20.18 | 0.252 | ~80M |
| AVSS (Zhou et al., 2022a) (ASPP) | 28.94 | - | ~180M |
| AVSS (Zhou et al., 2022a) (PVT-v2) | 29.77 | 0.352 | ~180M |
| Meta-Transformer | **31.33** | **0.387** | **86.5M** |

the performance of Meta-Transformer and existing methods on the AVSS dataset for audio-visual segmentation. The evaluation metrics reported in this task are mIou and F-score. In comparison, Meta-Transformer outperforms all other methods with the highest mIou of 31.33% and the highest

F-score of 0.387. It also stands out for its significantly lower parameter count, with only 86.5 million parameters compared to the approximate 80M to 180M parameters of other methods.

Meta-Transformer offers several advantages over other methods in the field.

- **Unified architecture**. It relieves modality-specific encoders and reduces computation by leveraging a unified encode to process both audio and images, resulting in a more efficient and streamlined process.

- **Faster convergence**. Thanks to the unified architecture for processing both audio and images, the encoder can deeply align the two modalities instead of only at the output end, which leads to faster convergence. Meta-Transformer only needs 4 training epochs to reach 31.33% of mIou.

- **Superior performance**. Meta-Transformer achieves a significant improvement of $10\%$ compared to other methods of a similar parameter scale.

- **Efficiency**. Despite its enhanced performance, Meta-Transformer achieves this with much fewer parameters, requiring only $1/3$ of the parameter amount, which makes forward and backward progress ease.

In summary, the benefits of employing the Meta-Transformer to deal with multi-modal tasks are appealing due to computational efficiency, rapid convergence, improved performance, and parameter efficiency. It reveals the significantly promising direction to apply Meta-Transformer to more multi-modal tasks.

## D   EXPERIMENTAL DETAILS

**Text understanding**. For text understanding evaluation, we employ the General Language Understanding Evaluation (GLUE) benchmark (Wang et al., 2018) which incorporates several different datasets, covering a wide range of natural language understanding tasks.

The comparison centers on paraphrasing, sentiment, duplication, inference, and answering tasks. When using frozen parameters pretrained on images, Meta-Transformer-B16$_F$ achieves scores of 54.6% in sentiment (SST-2), 81.1% in paraphrase (MRPC), 66.0% in duplication (QQP), 63.4% in inference (MNLI), and 56.3% in answering (QNLI) tasks.

**Image understanding**. 1) Classification: we conduct experiments on ImageNet-1K (Deng et al., 2009) which contains approximately 1.3 million images with 1000 categories. Following common practices (Wang et al., 2021b; Liu et al., 2021b; 2022c), base-scale models are trained for 300 epochs, while large models are pre-trained on ImageNet-22K (14.2 million images) for 90 epochs and fine-tuned on ImageNet-1K for another 20 epochs. 2) Object Detection: we conduct experiments on the MS COCO dataset (Lin et al., 2014) using Mask R-CNN (He et al., 2017) as the detector and training each model for 12 epochs. 3) Semantic Segmentation: we train the segmentation head UperNet (Xiao et al., 2018) on ADE20K (Zhou et al., 2017) for 160k iterations, providing a fair comparison with previous CNN-based and transformer-based backbones.

With the Meta-Transformer-B16$_F$ and Meta-Transformer-L14$_F$, achieving 69.3% and 75.3%, respectively. At the same time, when the pretrained parameters are further tuned, Meta-Transformer can outperform existing advanced methods.On object detection, Meta-Transformer-B16$_F$ and Meta-Transformer-L14$_F$ achieve APs of 31.7% and 43.5%, while Meta-Transformer-B16$_T$ and Meta-Transformer-L14$_T$ reach 46.4% and 56.3% AP, respectively. In semantic segmentation, the mIoUs for Meta-Transformer-B16$_F$ and Meta-Transformer-L14$_F$ are 33.4% and 41.2%, while Meta-Transformer-B16$_T$ and Meta-Transformer-L14$_T$ achieve 51.0% and 55.0%, respectively. In comparison, SwinV2-L/24[‡] outperforms the Meta-Transformer in both object detection (58.8% AP) and semantic segmentation (55.9% mIoU). These results highlight that Meta-Transformer demonstrates a competitive performance in various image understanding tasks even compared to Swin Transformer (Liu et al., 2021b) and InternImage.

**Infrared, X-Ray, and Hyperspectral data understanding**. We conduct experiments on infrared image, X-Ray scan, and hyperspectral data recognition with RegDB (Nguyen et al., 2017), Chest X-Ray (Rahman et al., 2020), and Indian Pine [1] datasets, respectively.

**Point cloud understanding**. 1) Classification: to assess the performance of Meta-Transformer in 3D object classification, we use the ModelNet-40 (Wu et al., 2015) benchmark, consisting of CAD models across 40 classes, with 9,843 training samples and 2,468 validation samples. 2) Semantic segmentation: to evaluate performance in 3D point cloud segmentation, we assess the model on both S3DIS (Armeni et al., 2016) and ShapeNetPart (Yi et al., 2016) datasets. The S3DIS dataset encompasses 6 large indoor areas and 13 semantic classes, comprising 271 rooms. The ShapeNetPart dataset includes 16,880 object models across 16 shape categories.

When pretrained on 2D data, Meta-Transformer-B16$_F$ demonstrates competitive performance, achieving an overall accuracy (OA) of 93.6% on ModelNet-40 with only 0.6M trainable parameters, which is comparable to the best-performing models. On the S3DIS Area-5 dataset, Meta-Transformer outperforms other methods with a mean IoU (mIoU) of 72.3% and a mean accuracy (mAcc) of 83.5%, using 2.3M parameters. Moreover, Meta-Transformer excels in the ShapeNetPart dataset, achieving the highest scores on both instances mIoU (mIoU$_I$) and category mIoU (mIoU$_C$) with 87.0% and 85.2%, respectively, using 2.3M parameters.

**Audio recognition**. For audio recognition, we utilize the Speech Commands V2 (Warden, 2018) dataset, which consists of 105,829 one-second recordings of 35 common speech commands. Meta-Transformer-B16T model exhibits a significantly higher accuracy of 97.0% when tuning the parameters, whereas the AST model only reaches an accuracy of 92.6%. When AST is pre-trained on ImageNet and supplemented with additional Knowledge Distillation (KD), it achieves an improved performance of 98.1%, but with a higher number of trainable parameters of 86.9M. SSAST models display accuracy scores ranging from 97.8% to 98.0% while requiring 89.3M parameters. These results highlight that the Meta-Transformer performs competitively in the audio domain, demonstrating its versatility and effectiveness across different fields.

**Video recognition**. For video understanding, we conduct experiments on the UCF101 (Soomro et al., 2012) dataset for action recognition, with more details presented in § B.1.

**Time-series forecasting**. For time-series forecasting, we conduct experiments on ETTh1 (Zhou et al., 2021), Traffic[2], Weather[3], and Exchange (Lai et al., 2018) datasets. We use the tokenizer of Autoformer (Wu et al., 2021).

**Graph understanding**. We conduct experiments on the PCQM4M-LSC dataset (Hu et al., 2021), which is a large-scale dataset consisting of 4.4 million organic molecules with up to 23 heavy atoms with their corresponding quantum-mechanical properties. With the target of predicting molecular properties using machine learning, it has plenty of applications in drug discovery, and material science.

**Tabular analysis**. We conduct experiments on adult and bank marketing from UCI repository [4]. We use the tokenizer of TabTransformer (Huang et al., 2020) to encode raw tabular data.

**IMU recognition**. To evaluate the ability of Meta-Transformer to understand the inertial motion systems, we conduct experiments of IMU sensor classification on the Ego4D (Grauman et al., 2022) dataset.

### D.1  ABLATION STUDY

we mainly conduct the ablation experiments, which are relevant to the depth of tuning transformer blocks, and pretraining on tokenizers as shown in Table 14 and Table 15.

---

[1] https://github.com/danfenghong/IEEE_TGRS_SpectralFormer/blob/main/data/IndianPine.mat

[2] https://pems.dot.ca.gov/

[3] https://www.bgc-jena.mpg.de/wetter/

[4] http://archive.ics.uci.edu/ml/

| Models | Pretrained Tokenizer | Modality | Performance (%) |
|---|---|---|---|
| Meta-Transformer-B16 | From Scratch | Video | 54.22 |
| Meta-Transformer-B16 | VideoMAE | Video | 57.11 |
| Meta-Transformer-B16 | From Scratch | Image | 85.42 |
| Meta-Transformer-B16 | MAE | Image | 85.93 |

Table 14: Ablation study on tokenizer components.

| Models | Transformer Depth | ImageNet-1K (%) |
|---|---|---|
| Meta-Transformer-B16 | 1 | 42.74 |
| Meta-Transformer-B16 | 2 | 58.91 |
| Meta-Transformer-B16 | 4 | 75.63 |
| Meta-Transformer-B16 | 8 | 83.98 |
| Meta-Transformer-B16 | 12 | 85.42 |

Table 15: Ablation study on fine-tuning transformer blocks.

Our code is built on open-source projects including MMClassification[5], MMDetection[6], MMsegmentation[7], OpenPoints[8], Time-Series-Library[9], Graphomer [10].

We sincerely thank their great contributions. More implementation details can be found in our source code.

# E LIMITATION

From the perspectives of complexity, methodology, and further application, the limitations of the Meta-Transformer are summarized as follows:

**Complexity**: Meta-Transformer requires $\mathcal{O}(n^2 \times D)$ computation dealing with token embeddings $[\boldsymbol{E}_1, \cdots, \boldsymbol{E}_n]$. High memory cost and heavy computation burden make it difficult to scale up.

**Methodology**: Compared with Axial Attention mechanism in TimeSformer (Bertasius et al., 2021) and Graphormer (Ying et al., 2021), Meta-Transformer lacks temporal and structural awareness. This limitation may affect the overall performance of Meta-Transformer in tasks where temporal and structural modeling plays a critical role, such as video understanding, visual tracking, or social network prediction.

**Application**: Meta-Transformer primarily delivers its advantages in multimodal perception. It's still unknown about its ability for cross-modal generation. We will work on this in the future.

# F FURTHER IMPACT DISCUSSION

## F.1 MODALITY-FREE PERCEPTION

We hope that Meta-Transformer can introduce new insight into both multi-modal learning and multi-modal generation fields. Meta-Transformer enables the usage of a shared encoder to encode diverse modalities, e.g. natural language, 2D images, 3D point clouds, as well as audio spectrograms., and project them into a shared representation space. This naturally reduces the modality gap across

---

[5]https://github.com/open-mmlab/mmpretrain/tree/mmcls-1.x

[6]https://github.com/open-mmlab/mmdetection

[7]https://github.com/open-mmlab/mmsegmentation

[8]https://github.com/guochengqian/openpoints

[9]https://github.com/thuml/Time-Series-Library

[10]https://github.com/microsoft/Graphormer

modalities and mitigates the burden of cross-modal alignment. In addition, Meta-Transformer removes the need for paired training data (such as image-text pairs), thus endowing multi-modal learning with more training flexibility.

## F.2 APPLICATION PROSPECTS

We investigate the application of Meta-Transformer on a wide range of modalities including RGB images, text, point clouds, video understanding, remote sensing (hyper-spectral images), nighttime surveillance (infrared images), and medical analysis (X-Ray images).

**In video understanding**, Meta-Transformer reveals the potential of enhancing the analysis and interpretation of videos by integrating information from text, audio, and image with the shared encoder. This benefits tasks such as action recognition, event detection, and video summarization. Meta-Transformer's capability to handle video-related modalities paves the way for improved video understanding applications in areas like video surveillance, video indexing, and content-based video retrieval.

**In hyperspectral imaging for remote sensing**, Meta-Transformer enables the analysis and understanding of hyperspectral data by extracting high-level semantic features. It enhances tasks such as classification, target detection, and land cover mapping, improving the accuracy and efficiency of remote sensing applications. The ability to process hyperspectral images using Meta-Transformer opens doors for advancements in environmental monitoring, agriculture, urban planning, and disaster management.

**In medical applications**, particularly X-ray image analysis, Meta-Transformer offers a promising approach to improving diagnostic accuracy and efficiency with multi-modal information. It can effectively capture and fuse information from X-ray images, clinical data, and other modalities to aid in disease detection, anomaly identification, and treatment planning by leveraging its unified learning framework. Meta-Transformer's capability to handle multi-modal data enhances the potential for more accurate and comprehensive medical imaging analysis, leading to better patient care and outcomes.

**For infrared images used in nighttime recognition and surveillance**, Meta-Transformer's ability to process infrared data helps extract crucial information for object detection, tracking, and recognition in low-light conditions, which opens an avenue for advancements in nighttime surveillance, security systems, and autonomous navigation in challenging environments with the cooperation between infrared cameras with RGB cameras.

## F.3 CONCLUSION

In summary, we think that the ability of Meta-Transformer to unify multi-modal learning comes from that *neural network architectures can learn modality-invariant patterns*. The architecture of Meta-Transformer illustrates the advantages of length-variable token embeddings in multi-modal learning, which provides flexible but unified forms of multi-modal semantics. Then it's time to think about designing algorithms to train networks that generalize on *unseen* modalities. Meanwhile, it's also intriguing to design the architecture of a unified multi-modal decoder, which can decode representations into any form of a specific modality.

Although Meta-Transformer presents a surprising performance and shows a new promising direction in multi-modal perception, we are not sure whether the proposed architectures are also effective in generative tasks. And it remains mysterious how to develop modality-invariant generative models. We hope that this can inspire future research.

