# OpenReview forum: "Meta-Transformer: A Unified Framework for Multimodal Learning"
_ICLR.cc/2024/Conference — ICLR 2024 Conference Withdrawn Submission_

### Official Review · Reviewer_NoxF · 2023-10-20

**Soundness:** 3 good
**Presentation:** 3 good
**Contribution:** 2 fair
**Rating:** 6
**Confidence:** 3

**Summary:**

This paper explores the potential of plain transformers for unified multimodal learning, highlighting a promising trend toward developing unified multimodal intelligence with a transformer backbone. To some extent, this paper supports the dominant position of transformers in next-generation networks. Importantly, CNNs and MLPs are not left behind. They play essential roles in data tokenization and representation projection. This process exemplifies the law of succession in neural networks and the ongoing evolution of artificial intelligence.

**Strengths:**

+ For multimodal research, the paper proposes a novel framework, Meta-Transformer, which utilizes a unified encoder to simultaneously extract representations from multiple modalities with the same set of parameters.
+ For multimodal network design, the paper comprehensively examines the functions of transformer components (e.g. embeddings, tokenization) and encoders in processing various modalities. Meta-Transformer provides valuable insights and sparks a promising new direction in developing a modality-agnostic foundation model capable of unifying all modalities.
+ Experimentally, Meta-Transformer achieves outstanding performance on various datasets spanning 12 modalities and excels in multimodal understanding, which validates the further potential of Meta-Transformer for unified multimodal learning.

**Weaknesses:**

- The paper has very beautiful figures and conducts very hard work for 12 modalities, datasets, tasks, loss functions, heads. One of the major weaknesses of the paper is that, the novelty might be not enough for a top conference. There is no much innovation in the data to sequence tokenization. All the tokenization including patch embedding, word embedding, etc., are existing strategies. The framework of the method is a widely used ViT. I acknowledge the hard work of datasets and experiments authors conduct. It would be better if novel methods or architectures could be proposed.

- Although there are extensive experiments conducted in the paper, the results are worse than the state-of-the-art approaches on GLUE, and other datasets. For the classification task, the best result is 91.1% in https://paperswithcode.com/sota/image-classification-on-imagenet and the paper reports 88.1%. It could improve the paper's quality a lot if there could achieve better than state-of-the-art results.

**Questions:**

Please read weaknesses

---

### Official Review · Reviewer_UnBq · 2023-10-31

**Soundness:** 2 fair
**Presentation:** 3 good
**Contribution:** 2 fair
**Rating:** 3
**Confidence:** 4

**Summary:**

This work extends the original transformer to train 12 different data modalities across image, text, audio, video, point cloud, the authors termed it as meta-transformer. The main hypothesis is that there is an effective parameter space that captures semantic meaning from different modalities and can be used to process data from that modality. Beyond a unified encoder, the authors designed modality-specific data processing layers and heads. The authors conducted extensive experiments to: (1) compare the two versions of the proposed architectures, namely, the frozen encoder and the fine-tuned encoder; and (2) compare them to the other recently proposed baselines on benchmark datasets.

**Strengths:**

1. The work addresses an important and interesting topic in multimodal learning and tries to cover up to 12 modalities. Additionally, I think it is well structured and easy to follow.

2. The authors conducted very extensive experiments and comparisons.

**Weaknesses:**

1. It is a bit difficult for me to draw the conclusion that the proposed method performs better than other baselines. For example, in Table 3, Table 5, Table 8, we can see a clear performance gap as compared to other baselines.

2. Another concern is about novelty. I feel the technical novelty is limited; a similar concept has been explored widely since [1]. The difference is mainly about the shared component.

[1] Ngiam, Jiquan, et al. "Multimodal deep learning." Proceedings of the 28th international conference on machine learning (ICML-11). 2011.

**Questions:**

1. The proposed method enables training without paired data. I am wondering how this method could capture and leverage the semantic correlations across modalities.

2. On some tasks, meta-transformers perform way worse than baselines (e.g., Tables 3, 5, and 11), what could be the reasons for this?

3. This work involves multiple metrics across tasks; it would be good to have a table in the appendix to summarize the tasks and their corresponding metrics.

---

### Official Review · Reviewer_i7Pr · 2023-10-31

**Soundness:** 3 good
**Presentation:** 3 good
**Contribution:** 3 good
**Rating:** 6
**Confidence:** 4

**Summary:**

This paper introduces a new multimodal framework based on a frozen large language model, capable of handling and managing various modalities. For training this system, the authors have collected a substantial amount of data involving audio and 3D questions. Additionally, they propose a new evaluation task designed to test the model's performance in cross-modal reasoning.

**Strengths:**

1. The paper is well-written and presents its ideas clearly.
2. The proposed Meta-Transformer framework demonstrates significant innovation and practicality in handling multimodal learning, especially with unpaired data.
3. The results provided across multiple benchmarks validate the effectiveness of your approach and cover a wide range of applications, providing evidence of the method's broad applicability and robustness.
4. The performance of Meta-Transformer on cross-modal retrieval, referring segmentation, and grounding tasks offers valuable contributions to the field of multimodal understanding.

**Weaknesses:**

1.	The model has achieved commendable results, but I believe that further scaling up the model could potentially yield even more intriguing outcomes.

2.	It is noted that the base model parameters are frozen during the training of different tasks. Therefore, most of the model's capabilities actually stem from contrastive learning between images and text. I think this approach to model training is still quite distant from achieving a truly universal model, as contrastive learning largely focuses on aligning modalities.

3.	Could the authors provide results for when the backbone is trainable? I understand that if the model is unfrozen, different tasks may compete with each other, potentially leading to poorer performance with smaller model sizes. However, I believe such findings wouldn't detract from the quality of the paper and would be more meaningful for the community.

**Questions:**

Please see weaknesses.

---

### Official Review · Reviewer_xeZr · 2023-11-02

**Soundness:** 1 poor
**Presentation:** 2 fair
**Contribution:** 2 fair
**Rating:** 3
**Confidence:** 4

**Summary:**

The proposed Meta-Transformer addresses the challenge of designing unified networks for multimodal learning, specifically bridging inherent gaps among varied modalities like natural language, images, audio, and more. Unlike traditional approaches, Meta-Transformer maps raw inputs from these modalities into a shared token space using a frozen encoder, even without paired multimodal training data. Comprising a unified data tokenizer, a modality-shared encoder, and task-specific heads, it stands out as the first to achieve unified learning across 12 modalities. Experiments indicate its broad applicability in various tasks and superior performance in multimodal understanding.

**Strengths:**

1. It is interesting to see a model handling 12 modalities.
2. The proposed idea is straightforward.
3. The paper is presented overall clearly.

**Weaknesses:**

1. Despite the success of one model handling multiple modalities, the insight provided in rather limited. There are many important questions that are not really answered.

a. Why using the meta-transformer in this pretrained manner? How about other pretraining manners on images? How about pretrained transformer in other modalities like text?

b. The conclusion also touches a claim that transformer is the future universal architecture. However, other architectures are not really validated. On the computer vision side, recent CNN or even MLP based method can achieve comparable or even better performance than vanilla transformer. [a]

c. It is shown that in many of the tasks, the performance is not that really better than SOTA. It is really unclear whether the usage of the multi-modal unified model is better or not. It is important to understand which modality is benefiting from the unified pretraining, why this modality can benefit, and which modality is more helpful to the other modality. The selection of the modalities is also arbitrary.

d. The supplementary also highlights the convergence. It is also unclear whether this really comes from joint training. It is really necessary to establish a comparable baseline to really analyze the effect of pretrained model usage, the data, the modality and the specific module design.

[a]. MetaFormer Is Actually What You Need for Vision

**Questions:**

Please check weakness for details.

---

### Official Review · Reviewer_7xkR · 2023-11-08

**Soundness:** 3 good
**Presentation:** 3 good
**Contribution:** 2 fair
**Rating:** 5
**Confidence:** 3

**Summary:**

Due to the information gap between different modalities, there is no unified model for multimodal problems. This paper addresses this issue by exploring the advantages of the Transformer architecture in multimodal feature learning. To map the original data to a shared token space for feature learning, this paper employs different tokenization methods for different modalities, and uses a pretrained and frozen Transformer encoder as the backbone. Task-specific heads are proposed for different tasks. Different from early work, paired data is not used in the feature learning process, making it the first framework to attempt feature learning on 12 different modalities. Experimental results show that the Meta-Transformer can handle fundamental single-modality cognitive task to some extent, demonstrating its potential in real-world applications and data mining. The Transformer architecture exhibits certain advantages in multimodal learning.

**Strengths:**

1 To address the challenge of learning from multiple modalities, the authors propose a unified pipeline that includes a modality-specialist for data-to-sequence tokenization, a modality-shared encoder for extracting representations across modalities, and task-specific heads for downstream tasks. This provides a comprehensive solution for multimodal learning.

2 To showcase the capabilities of the Transformer in multimodal learning, a wide range of modalities and tasks were utilized for training, highlighting the significant potential of Transformer architecture in multimodal learning and achieving acceptable results.

3 While the performance on individual modality tasks may be lacking, the advantages of the model become evident in multimodal tasks, surpassing state-of-the-art methods.

**Weaknesses:**

1 There appears to be a discrepancy in the description of the tokenization process. In Figure 3, it shows the use of a 1x1 convolution for feature dimension mapping during tokenization. However, on page 6, in the first line, it mentions the use of CLIP for learning word embeddings. This seems to be conflicting information. It's important to clarify and ensure consistency in the tokenization process described in the paper.

2 I think using the Visual Transformer (ViT) for encoder in pretraining and freezing parameters during other task learning is not an ideal approach. This approach assumes that high-dimensional visual information is necessarily the intersection of information from multiple modalities, as suggested in Section 3.1 of the paper.

3 The description of downstream tasks in the paper appears to be insufficient. It doesn't provide a clear explanation of how MLPs (Multi-Layer Perceptrons) are used to address different tasks, and it doesn't clarify whether MLPs are always the suitable approach for different tasks.

4 In the fourth section of the paper, where F and T represent freezing and fine-tuning of the encoder, respectively, there seems to be a need for clarification regarding how multitask learning is conducted under the freezing approach and how fine-tuning is applied to different tasks.

5 The experimental section of the paper appears to have room for improvement. Despite conducting numerous experiments and showcasing the performance of MetaFormer on individual tasks, the results are not as promising as expected. It seems that the model's introduction does not significantly enhance performance on single tasks, and it may not effectively leverage the information across multiple modalities to enhance the learning of a particular modality. Additionally, it's important to provide more comprehensive experimental analysis, particularly for cases where only simple demonstrations of results are presented. The paper should delve into the reasons for the observed performance, potential limitations, and insights gained from the experiments. This can help provide a clearer understanding of the model's strengths and weaknesses.

6 Formatting issues are crucial as they can directly impact the readability and professionalism of the paper. On page 8, particularly in Table 8, both parts a and b of the table exhibit misalignment and overall disarray in the formatting. It is imperative to rectify the formatting to ensure readability. I recommend that the authors reformat the table, aligning the columns properly to enhance the quality and readability of the paper.

7 Writing issues, such as minor spacing errors, can significantly impact the quality of a manuscript. In the third line on page 9, it reads, "Table 8b provides the comparison between different methods for tabular data understanding." There is indeed a missing space, and the correct phrasing should be: "Table 8b provides the comparison between different methods for tabular data understanding." Please be sure to thoroughly proofread your manuscript to correct such spelling and formatting errors.

**Questions:**

Please refer to the weaknesses above.